# Industrial-Scale Technology for Molybdic Acid Production from Waste Petrochemical Catalysts

**DOI:** 10.3390/ma16175762

**Published:** 2023-08-23

**Authors:** Katarzyna Leszczyńska-Sejda, Piotr Dydo, Ewa Szydłowska-Braszak

**Affiliations:** 1Łukasiewicz Research Network, Institute of Non-Ferrous Metals, 44-121 Gliwice, Poland; katarzyna.leszczynska-sejda@imn.lukasiewicz.gov.pl; 2Faculty of Chemistry, Silesian University of Technology, 44-100 Gliwice, Poland; piotr.dydo@polsl.pl

**Keywords:** metal recovery, molybdenum, spent catalyst, hydrodesulfurization

## Abstract

The article describes the technology of molybdic acid recovery from spent petrochemical catalysts (HDS) developed and implemented in industrial activity. HDS catalysts contain molybdenum in the form of MoO_3_ and are used for the hydrodesulfurization of petroleum products. After deactivation, due to the impurities content in the form of sulfur, carbon and heavy metals, they constitute hazardous waste and, at the same time, a valuable source of the Mo element, recognized as a critical raw material. The presented technology allows the recovery of molybdic acid with a yield of min. 81%, and the product contains min. 95% H_2_MoO_4_. The technology consisted of oxidizing roasting of the spent catalyst, then leaching molybdenum trioxide with aqueous NaOH to produce water-soluble sodium molybdate (Na_2_MoO_4_), and finally precipitation of molybdenum using aqueous HCl, as molybdic acid (H_2_MoO_4_). Industrial-scale testing proved that the technology could recover Mo from the catalyst and convert it into marketable molybdic acid. This proves that the technology can be effectively used to preserve molybdenum.

## 1. Introduction

Molybdenum (Mo), Sb, Bi, B, Cu, Au, Re, and Zn belong to the geologically rarest elements group [1]. This high-melting metal is commonly applied to obtain ferrous or non-ferrous alloys used to produce parts for airplanes and industrial engines and components of nuclear reactors [2,3,4]. As a compound, molybdenum’s primary applications involve its use as an active component of hydrodesulfurization catalysts for treating crude oils [5,6,7]. Due to these applications, molybdenum has been classified as a critical raw element essential for global conversion to fossil fuel-free energy production [2,8]. It should be mentioned that international Mo production rates has increased more than 20-fold during the last 50 years: from 14,500 tons in 1950 to 300,000 tons in 2021 [2,9]. At the current mining rate of molybdenum minerals, the documented geological deposits of molybdenum will be exhausted during the next 50–100 years [1]. To conserve molybdenum for future generations, it is postulated that the molybdenum recycling rate should be increased from the current 20% to around 80% with a simultaneous reduction in the mining rate of molybdenum ores from nature of about 47% [1,2].

Spent hydrodesulfurization (HDS) catalysts, produced by the petrochemical industry at a high annual rate of 150,000 to 170,000 tons [6,10,11], constitute hazardous wastes [12,13] which contain: molybdenum mainly as molybdenum disulfide (MoS_2_), other sulfur compounds or elemental sulfur, coke, hydrocarbons, and combinations of other heavy metals (V, Ni or Co) along with alumina (Al_2_O_3_) carrier materials [14]. Their expected lifetime varies from 3 to 7 years, after which, despite cyclic regeneration, they lose their catalytic properties and become hazardous wastes [15,16]. Because of their high molybdenum content, spent HDS catalysts might constitute a valuable source of secondary molybdenum from which molybdenum metal or its compounds can be recycled and thus conserved [1,5,6,17,18].

The existing literature describes many methods for molybdenum recovery from spent HDS catalysts. Those methods are primarily based on high-temperature roasting and leaching treatment of the spent catalysts, leading to molybdenum recovery as molybdic acid (H_2_MoO_4_), molybdenum trioxide (MoO_3_), or molybdate salts next to compounds of other metals. Typically, roasting treatment of spent HDS catalyst involves oxidizing roasting at the temperature range from 500 to 1000 °C in the presence of air or oxygen, aimed at the oxidation of MoS_2_ to MoO_3_ along with oxidation of other metal sulfides, sulfur, coke, and other hydrocarbons [19,20,21]. In addition, oxidizing roasting may be conducted in the presence of sodium or potassium compounds, such as carbonate, hydroxide, chloride, or bisulfate, in which MoS_2_ is converted into water-soluble molybdate salts, e.g., Na_2_MoO_4_ [11,19]. Furthermore, molybdenum compounds can be leached indirectly from the spent HDS catalysts by leaching the pre-roasted catalyst with water, acidic, alkaline, or ammonia solutions [11,19,22,23,24,25]. Direct leaching of unroasted spent catalysts with acids, bases, or ammonia at elevated temperatures, also examined extensively, was quite often conducted in the presence of an oxidant such as oxygen, hydrogen peroxide, ferric chloride, nitric acid, sodium hypochlorite or sodium chlorate aimed at the oxidation of MoS_2_ to hexavalent oxo-compounds of molybdenum, such as molybdic acid or molybdates [3,6,7,19,23,26,27,28,29]. Direct or indirect leaching of the spent HDS catalysts with acidic effluents, such as aqueous HCl, H_2_SO_4_, HNO_3_, or low molecular weight carboxylic acids, typically results in the dissolution of molybdenum along with other metals: vanadium, nickel, cobalt, and at least partially, aluminum [30,31].

Consequently, these metal compounds have to be separated from each other by selective precipitation (e.g., as sulfides), solvent extraction, or ion exchange [32,33,34,35,36,37]. In turn, leaching of roasted or unroasted spent HDS catalysts with ammonium buffers (ammonia–ammonium salt solutions) results in the dissolution of molybdenum as ammonium molybdate, along with vanadium as ammonium metavanadate and nickel or cobalt as water-soluble ammine complexes, but leaves aluminum as an insoluble solid residue of Al_2_O_3_ [11]. The highest selectivity in molybdenum recovery can be achieved with direct or indirect leaching of molybdenum with basic compounds of sodium or potassium, in which treatment of the catalyst or pre-roasted catalyst with hydroxide or carbonate solutions results in effective dissolution of molybdenum and vanadium as molybdate and metavanadate, respectively, with the limited dissolution of aluminum, the majority of which remains as insoluble Al_2_O_3_, along with nickel and cobalt which remain solid as hydroxides or oxides. Similar effects can be achieved by mixing the above caustic compounds or sodium chloride with the catalyst, roasting, and subsequently leaching molybdenum and vanadium with water or other aqueous media. However, this type of roasting needs to be carried out at relatively high temperatures, in the range from 600 to 900 °C, which quite often exceed the melting point for the caustics and result in the fusion of the catalyst with the reagents [11,19,22,24,25,38]. In most cases, molybdenum, after more or less extensive purification of the leachate, is precipitated in the form of ammonium molybdate or polymolybdate, which needs to be further calcined at around 450 °C to form MoO_3_ and produce NH_3_ vapors, which needs to be scrubbed from the gaseous phase [24,34,39,40]. Alternatively, molybdenum can be precipitated as molybdic acid by neutralizing the leachate with acids [41]. It should be mentioned that as an alternative to the above-described methods, molybdenum can be removed from the roasted catalyst as MoO_3_ by sublimation at relatively high temperatures, which consumes a lot of energy [15] or in a chlorination-roasting process, as MoCl_6_ along with AlCl_3_ which, however, consumes expensive and corrosive chlorine [19,22].

In this study, the technology for industrial-scale recovery of molybdenum in the form of molybdic acid from spent HDS catalysts was developed. The examined HDS catalyst contained molybdenum, nickel, and cobalt compounds supported on alumina. A three-stage process was proposed based on the above-presented literature survey and catalyst composition. In the first stage, the untreated spent catalyst was subjected to oxidizing roasting with air, aimed at the oxidation of molybdenum compounds to MoO_3_ and removal of the majority of the sulfur. In the second stage, molybdenum was leached from the roasted catalyst with NaOH solution. At the same time, the majority of nickel, cobalt, and alumina remained solid and was subjected to further treatment to recover Ni or Co. This was followed by neutralization of the molybdenum-rich leachate with HCl and precipitation of molybdenum in the form of molybdic acid (H_2_MoO_4_) at a pH of around 0 in the third stage. The scheme for the technology applied is presented in Figure 1. Alkaline leaching of the roasted HDS catalyst with NaOH was adopted due to its high selectivity towards molybdenum and the relatively low price of the reagent. Oxidizing roasting with air as a first stage of treatment was selected as an alternative to the roasting of the catalyst mixed with solid NaOH, since simple roasting can be conducted at relatively low temperatures and prevents the necessity of grinding the products of fusion of the catalyst with NaOH, which would constitute an energy-consuming operation. Precipitation of molybdenum as molybdic acid, in place of more commonly reported precipitation of molybdenum as ammonium molybdate, was applied because of the lower solubility of molybdic acid, hence the relative easiness of its separation. In addition, precipitation of molybdenum as molybdic acid rather than as ammonium molybdates lets us avoid the necessity for calcination of ammonium salts in order to convert them to MoO_3_, which requires the relatively high temperature of around 450 °C and would constitute another energy-demanding operation and introduce the need for removal of ammonia or other gaseous calcination products from the vapor phase by scrubbing. Moreover, molybdic acid constitutes the suitable starting point for reduction to the metallic form or conversion to other molybdenum compounds. When needed, molybdic acid can be easily dehydrated to MoO_3_ by heating at just above 100 °C. The dried molybdic acid produced therein constitutes a tradeable product. In this work, the optimum conditions for each stage of the process mentioned above were evaluated in the laboratory and then tested on an industrial scale. Based on the obtained results, the effectiveness of the examined technology in molybdenum recycling and conservation are discussed.

## 2. Materials and Methods

### 2.1. Oxidizing Roasting Experiments

The optimum conditions for the molybdic acid production process, presented schematically in Figure 1, were first evaluated in the laboratory. The effects of temperature in the range from 400 to 700 °C, and time in the range of 2 to 10 h, on the effectiveness of sulfur removal and molybdenum recovery during the roasting of spent HDS catalyst were first evaluated. To analyze the obtained data statistically, an experimental protocol was designed for response surface type evaluation of data using the D-optimal method included in Design Expert 6.0.8 (Stat-Ease, Minneapolis, MN, USA) software. A total of 12 experiments were performed separately.

Laboratory-scale roasting tests were performed using a muffle-type electric furnace (NEOTERM KXP 4, OHAUS, Greifensee, Switzerland). In each case, a 10.0 g sample of the spent catalyst was weighed to a ceramic crucible of 45 mL volume and introduced to the furnace preheated to the examined temperature. Upon completion of the experiments, partial molybdenum recovery (*R_Mo,or_*) of the oxidation stage was calculated using the following formula:(1)$$R_{Mo,or}=\frac{m_{f}\times w_{Mo,f}}{m_{i}\times w_{Mo,i}}\times 100\%$$
where *m_f_* is the final mass of the roasted catalyst, *m_i_* is the initial mass of the raw catalyst, *w_Mo,f_* is the mass fraction of the molybdenum in the roasted catalyst, and *w_Mo,i_* is the mass fraction of the molybdenum in the raw catalyst. Sulfur removal (desulfurization) efficiency (*SR*) was calculated based on the following formula:(2)$$SR=\frac{m_{i}{\times w}_{S,i}-m_{f}{\times w}_{S,f}}{m_{i}{\times w}_{S,i}}\times 100\%$$
where *w_S,f_* is the mass fraction of sulfur in the roasted catalyst and *w_S,i_* is the mass fraction of sulfur in the raw catalyst.

Molybdenum mass fractions in this work were determined by the fusion of catalyst sample with potassium carbonate at 1000 °C for 30 min, dissolution of the fused product in deionized water, and determination of the molybdenum content using Atomic Absorption Spectrometry (iCE 3300 AA Spectrometer, Thermo Fisher Scientific, Waltham, MA, USA) at 313.2 and 320.2 nm wavelengths. The sulfur mass fraction was determined based on the mass of BaSO_4_ obtained by oxidizing the catalyst sample with sodium peroxide and sodium carbonate at 800 °C. Then the cooled product was dissolved in deionized water and HCl, and finally, barium chloride was added to precipitate the sulfate produced by the sulfur oxidation.

Industrial-scale roasting experiments were performed using a rotary kiln heated from the inside by a natural gas burner operated with excess air to ensure oxidizing conditions. The kiln was made of a 3.5 m long and 1.67 m in diameter steel drum lined from the inside with refractory materials for high-temperature insulation. The oxidizing roasting tests were performed in approx. 2000 kg batches at a kiln rotation speed of 15.6/h. The process temperature was controlled by the use of thermocouples positioned inside the roasting compartment. Oxidation tests were performed at the oxygen excess parameter, defined as the ratio between the actual amount of oxygen in the kiln related to the amount of oxygen required for complete methane combustion, equal to 1.75.

Powder X-ray diffraction (XRD) patterns of the catalyst samples and the molybdic acid produced in this work were acquired using an XRD-3003TT (Seifert, Radevormwald, Germany) diffractometer equipped with a long-line focus Cu X-ray tube and scintillation detector. To identify crystalline phases present in the samples, the collected XRD patterns were compared with the ICDD-PDF-2 database (International Centre for Diffraction Data, Delaware County, PA, USA).

### 2.2. Leaching Experiments

The effectiveness of molybdenum recovery from the roasted catalyst by leaching with NaOH solutions was evaluated in the laboratory. The effects of time in the range of 1 to 8 h, NaOH concentration in the range of 5 to 20%, temperature (20–80 °C), and liquid volume/solid mass (L/S, dm^3^/kg) ratio (3–7) on molybdenum and aluminum recovery and molybdenum removal selectivity were analyzed. A set of 25 experiments was designed to evaluate those effects statistically based on a D-optimal plan.

Every experiment was prepared by weighing a 50.0 g sample of the roasted catalyst to a 500 mL glass beaker, adding the required volume of NaOH solution, and heating up with a heating plate (CHEM-LAND HP550-S, “CHEMLAND” Technical and Commercial Enterprise, Stargard, Poland) for the required time. During the experiments, solutions were continuously stirred at 160 rpm with an electric stirrer (CHEM-LAND OS20-pro, Stargard, Poland). Upon completion of the experiment, solid residues were filtered using filter paper under vacuum (700 mbar, Rocker 801 vacuum pump, Sterlitech, Auburn, WA, USA) and washed with deionized water up to a total volume of the liquid sample of 500 mL. The molybdenum and aluminum content in these solutions were then determined using Atomic Absorption Spectrometry. The values of the partial recovery of molybdenum and aluminum of the leaching step, which describe the efficiency of this stage of the process and the Mo recovery selectivity index, were calculated according to the following formulas:(3)$$R_{Mo,l}=\frac{C_{Mo}\times V}{m_{f}\times w_{Mo,f}}\times 100\%$$
(4)$$R_{Al,l}=\frac{C_{Al}\times V}{m_{f}\times w_{Al,f}}\times 100\%$$
(5)$$S_{l}=\frac{C_{Mo}V}{m_{f}\times (w_{Mo,f}+w_{Al,f})}\times 100\%$$
where *C_Mo_* is the molybdenum content in the leachate (g/L); *C_Al_* is the aluminum content of the leachate (g/L); *V* is the volume of the leachate (L); *m_f_* is the mass of the roasted catalyst; *w_Mo,f_* is the mass fraction of the molybdenum in the roasted catalyst; and *w_Al,f_* is the mass fraction of the aluminum in the roasted catalyst.

Industrial-scale experiments on the leaching of molybdenum from roasted catalysts were performed in a cylindrical polypropylene stirred tank of 16.8 m^3^ volume, 3.2 m diameter, and 2.2 m height. The tank was equipped with a 70 cm wide stirrer rotated by a 27 kW electric engine operated at 15.0 rpm. During experiments, the temperature of the reaction mixture was maintained in the range of 50–65 °C by using overheated steam. Experiments were performed in five batches prepared by filling up the tank with the required volume of water to which solid NaOH was dosed until complete solubilization. Then approx. 2500 kg of roasted catalyst was introduced to the reaction mixture at a rate of 350 kg/h. The effectiveness of leaching was controlled periodically, after every 8 h, by determination of the amount of molybdenum in the leachate using atomic absorption spectrometry. After roughly 30 h of operation, the molybdenum content in the leachate stabilized, and the leaching procedure was terminated. Upon completion of leaching, solid residues were separated from the leachate using a plate-and-frame filter press (PFKM-630, Montech, Łuszczów Drugi, Poland) equipped with a PPT 1969S (Protex, Turek, Poland) membrane of a surface area of 15 m^2^. The filtrate was collected and subjected to further processing to precipitate molybdic acid.

### 2.3. Precipitation Experiments

Laboratory-scale molybdic acid precipitation experiments were performed by pouring 50 mL of the leachate into 250 mL glass beakers. Then the leachate pH was adjusted to the desired level using concentrated HCl solution and stirred with a magnetic stirrer at around 160 rpm for 2 h to precipitate H_2_MoO_4_ completely. Upon completion of precipitation, yellow solid molybdic acid was filtered under vacuum using filtering paper, and the molybdenum content in the supernatant solution was determined by Atomic Absorption Spectrometry (AAS). Simultaneously, the solid product was dried at 60 °C for 2 h and analyzed for molybdenum content by AAS. Molybdenum recovery in the precipitation step (*R_Mo,p_*) and molybdic acid content in the precipitate (% H_2_MoO_4_) were then determined as partial recovery specific for the precipitation step, using the following equations:(6)$$R_{Mo,p}=\frac{C_{Mo,l}\times V_{l}-C_{Mo,f}\times V_{f}}{C_{Mo,l}\times V_{l}}\times 100\%$$
(7)$$\%H_{2}MoO_{4}=\frac{m_{Mo,p}\times \frac{M_{ac}}{M_{Mo}}}{m_{p}}$$
where C_Mo,l_ is the molybdenum content in the leachate (g/L), C_Mo,f_ is the molybdenum content in the filtrate (g/L), V_l_ is the volume of the leachate (L), V_f_ is the volume of the filtrate (L), m_Mo,p_ is the mass of molybdenum in the dried precipitate (g), m_p_ is the mass of the dried precipitate (g), M_ac_ is the molar mass of molybdic acid (161.95 g/mol), and M_Mo_ is the atomic mass of molybdenum (95.95 g/mol).

Laboratory experiments on the effect of the addition of chloride salts on the solubility of molybdic acid were performed by the addition of 1.00 g of analytical grade molybdic acid to 50 mL of the solution which contained tested salt (NaCl, KCl, or NH_4_Cl) and the adjustment of the pH to the value of 0.0 with concentrated HCl. The samples prepared were then shaken for 24 h using a laboratory shaker WL-2000 (JWElectronics, Poland), operated at a frequency of 120/min. After that period, solid molybdic acid was removed by filtration under a vacuum using filtering paper, and filtrate samples were analyzed for molybdenum content.

Industrial-scale precipitation tests were performed using 16.8 m^3^ tanks applied previously for leaching operations. At first, the previously prepared leachate was introduced into the tank. Then, the leachate was acidified with concentrated hydrochloric acid, and a concentrated ammonia solution was finally introduced. Precipitation was carried out for approx. 2 h with continuous stirring of the reaction mixture at 15 rpm. Upon completion, the solid yellow product was filtered using a filtering press applied previously for leaching. The filtrate samples were collected and tested for molybdenum content while the solid molybdic acid was washed by stirring with approx. 8 m^3^ of deionized water for 2 h and subsequent filtration with a filtering press. The washed precipitate was then fed at 150 kg/h into the 9 m long and 1.1 m in diameter continuous rotary drier, heated with natural gas, and dried at a temperature range from 50 to 100 °C. A yellow solid product was obtained in the process.

## 3. Results

The composition of the exemplary spent HDS catalyst investigated in the laboratory is summarized in Table 1. The catalyst was characterized by a relatively high molybdenum and sulfur content of 11.6% and 7.7%, respectively. It also contained significant amounts of aluminum, possibly as porous Al_2_O_3_ support, along with approx. 5% of nickel and 2% of cobalt. It should be mentioned that in molar terms that sulfur, present in spent HDS catalysts most likely as sulfide (e.g., MoS_2_) or elemental sulfur [11,42,43], was the second principal component of the catalyst, after aluminum. 

The XRD pattern of the catalyst, presented in Figure 2, indicated a low signal-to-background ratio, which suggests a strongly amorphous catalyst morphology, possibly due to the presence of amorphous alumina. However, small amounts of crystalline matter could be matched to the pattern based on the presence of the strongest diffraction lines specific for hexagonal molybdenum disulfide (molybdenite), MoS_2_ (ICDD PDF-2 #02-0132); synthetic (cubic) alumina, Al_2_O_3_ (ICDD PDF-2 #10-0425); and octahedral sulfur, S_8_ (ICDD PDF-2 #02-0324). No evidence of nickel or cobalt compounds was found due to their low concentration or the poor crystallinity of the catalyst sample. The XRD results, along with the elemental composition, indicated that molybdenum disulfide, MoS_2_, constitutes the primary molybdenum compound of the examined catalyst. Strong broadening of the alumina XRD lines indicated the presence of small and poorly developed crystallites of cubic Al_2_O_3_ only.

### 3.1. Results of Oxidizing Roasting

Due to the inertness of MoS_2_, molybdenum cannot be leached directly from the spent catalyst, thus the oxidizing roasting step needs to be applied, aiming at the oxidation of sulfur and molybdenum, represented by the following reaction scheme:2 MoS_2_(s) + 7 O_2_ → 2 MoO_3_(s) + 4 SO_2_(8)

Laboratory experiments on the effectiveness of oxidizing roasting at the temperature range from 400 to 700 °C and time from 2 to 10 h were performed following the experimental plan described in paragraph 2. The data obtained were analyzed statistically for molybdenum recovery (R_Mo,or_) and sulfur removal efficiency (SR) and are summarized in Table 2. There was a relatively high molybdenum recovery of 93.1 ± 3.4 as well as a high sulfur removal efficiency of 86 ± 13. The highest molybdenum recovery was 97.2%, and the highest sulfur removal efficiency was 99%. Analysis of the correlation matrix (Table 3) shows that the sulfur removal efficiency (SR) correlates strongly with reaction time (R = 0.85), while its correlation with temperature remains poor (R = 0.09) and not statistically significant. On the other hand, there was a strong and statistically significant negative correlation of molybdenum recovery (R_Mo,or_) with reaction temperature (R = −0.95) and a weak, but not statistically significant, correlation with reaction time (R = 0.09) was observed. The obtained data are plotted in Figure 3, where a substantial decrease in molybdenum recovery with roasting temperature can be observed, while roasting time shows only a slight asymptotic effect on this process parameter. In contrast, the effects of process parameters on desulfurization efficiency, summarized in Figure 4, show a substantial increase in desulfurization efficiency with time at only a slight increase in this parameter with temperature. The above observations suggest a kinetically controlled mechanism for the oxidation of sulfur compounds, which indicates that the rate of sulfur oxidation during oxidizing roasting in the examined temperature range should be controlled by roasting time and the amount of oxygen. The decrease in molybdenum recovery with temperature, in turn, can be explained by sublimation of MoO_3_, which is known to have a covalent character and sublimes easily [15,21,44,45] in an endothermic process (enthalpy of sublimation of MoO_3_ at around 500 °C is equal to around 400 kJ/mole and slightly decreases with temperature [46]. The enthalpy of MoO_3_ sublimation during oxidizing roasting of the examined catalyst may be significantly compensated for by exothermic oxidation of sulfur or its compounds, leading to spontaneous losses of MoO_3_ produced by oxidation, especially at high temperatures. The above effects led us to conclude that industrial-scale roasting should be carried out at low temperatures, in the approximate temperature range from 400 to 550 °C to ensure high molybdenum recovery, and for a time sufficient enough for almost complete oxidation of sulfur.

The above laboratory-scale findings on oxidizing roasting were validated on an industrial scale by roasting 25,600 kg of spent HDS catalyst in a rotary kiln fired with natural gas in batches of catalyst, each of approx. 2500 kg. The spent catalyst was characterized by an average molybdenum content of 10.6% and sulfur content of 8.6%. Roasting was performed at an average temperature of 550 °C for the time required to oxidize at least 85% of sulfur. After around 16 h of roasting, desulfurization efficiency reached approx. 86%, while a high molybdenum recovery of 95.7% was observed. The chemical composition of the raw and calcined catalyst is presented in Table 4. It should be mentioned that similar desulfurization efficiency and molybdenum recovery were achieved in the laboratory just after 3 h of experiments at around 550 °C; however, on the industrial scale, a much longer time was required, possibly because of the limited amount of oxygen passing through the kiln along with flue gas at λ = 1.75. Typically, on the industrial scale it took approx. 9 h to achieve 85% desulfurization at 95% molybdenum recovery at 540 °C and 20 m^3^/h natural gas with 300 m^3^/h air flow rates.

The XRD pattern of the roasted catalyst, presented in Figure 5, showed a higher degree of crystallinity (higher signal-to-background ratio) than the unroasted catalyst; however, XRD lines specific for cubic alumina still showed strong broadening, which indicates a poorly developed crystalline phase of Al_2_O_3_ for both: roasted and unroasted catalyst. The major crystalline forms of molybdenum matched to the roasted catalyst XRD pattern were: orthorhombic molybdenum oxide MoO_3_ (ICDD PDF-2 #35-0609), and orthorhombic aluminum molybdate Al_2_(MoO_3_)_3_ (ICDD PDF-2 #20-0034). Due to the amorphous nature of the catalyst, no evidence of nickel or cobalt compounds was found.

### 3.2. Results of Alkaline Leaching

As shown in Figure 1, the second step of the examined technology involves leaching of the molybdenum from the roasted catalyst using NaOH solutions. During this stage, solid MoO_3_ and Al_2_(MoO_3_)_3_ formed during oxidizing roasting are subject to a reaction with aqueous NaOH with the formation of sodium molybdate (Na_2_MoO_4_), which dissolves in the leachate. Therefore, the general reaction schemes for the molybdenum leaching process can be presented as follows:MoO_3(s)_ + 2 NaOH_(aq)_ → Na_2_MoO_4(aq)_ + 2 H_2_O(9)
Al_2_(MoO_3_)_3_ + 6 NaOH_(aq)_ → 3 Na_2_MoO_4(aq)_ + 2 Al(OH)_3_(10)

During this stage of operation, it is commonly reported that significant amounts of aluminum can be leached along with molybdenum [5,47,48] due to the amphoteric character of the aluminum oxide (or hydroxide) and formation of water-soluble aluminate complexes. The aluminum leaching reactions can be summarized as follows:Al_2_O_3(s)_ + 3 H_2_O + 2 NaOH_(aq)_ → 2 Na[Al(OH)_4_]_(aq)_(11)
Al(OH)_3(s)_ + NaOH_(aq)_ → Na[Al(OH)_4_]_(aq)_(12)

This dissolved aluminate can contaminate the leachate and create severe limitations at the molybdic acid precipitation step regarding the of amount of HCl required for neutralization and contamination of the product. Therefore, alkaline leaching stage conditions have to be optimized not only to maximize molybdenum recovery (*R_Mo,l_*) but also to limit recovery of other metals or to maximize overall molybdenum leaching selectivity (*S_l_*), defined as the ratio between the amounts of molybdenum that has been solubilized compared to the amounts of all the metals that have been solubilized (see Equation (5)). In contrast to aluminum, other metals potentially present in the spent HDS catalysts, such as nickel or cobalt, are commonly reported to resist alkaline leaching [49] due to the fact, that their basic oxides or hydroxides do not dissolve in alkaline solutions.

Laboratory optimization of the effectiveness of leaching of molybdenum using NaOH solutions was performed using samples of catalyst roasted on an industrial scale. In these experiments, the effects of leaching time, temperature, NaOH concentration, and L/S ratio were investigated by performing experiments described in more detail in the experimental section of this work. Statistical analysis of the data collected, shown in Table 5, indicated that molybdenum could be leached from the roasted catalyst at high molybdenum recovery and selectivity values of 83.1 ± 9.4% and 85.6 ± 8.0, respectively. The maximum molybdenum recovery equaled to 94.7% while the maximum molybdenum selectivity was 99.6%. At the same time, the aluminum recoveries observed were low: the average value equaled to 15.3 ± 9.9% with a minimum of just 0.3%. The data collected were analyzed further statistically in terms of the correlation of variables (Table 6) and plotted in Figure 6 and Figure 7. This analysis showed that recovery of molybdenum and aluminum does not depend upon reaction time or L/S ratio in a statistically significant manner. However, there is a very strong and statistically significant positive correlation between recoveries of both molybdenum and aluminum with leaching solution NaOH concentration, indicating that the effectiveness of solubilization of both metals increases with NaOH content, obviously as a result of reactions (9) and (10). In the case of aluminum, its recovery was also positively correlated with temperature, which seems to promote reaction (11) and possibly reaction (12). The above correlations indicate that to achieve high molybdenum recovery from the roasted catalyst tested herein, alkaline leaching should be carried out at as high an NaOH concentration as possible. However, the increase in NaOH content will be accompanied by an increase in aluminum solubilization rate, an effect that is widely reported in the existing literature [49,50,51]. Such an increase in aluminum solubilization rate manifested itself herein as a statistically significant correlation between molybdenum recovery and aluminum recovery. Further examination of the experimental data, summarized in Figure 6a–c, indicated that relatively high molybdenum recovery levels of more than 90% of the leaching step could be achieved with an NaOH content of approx. 15% and below, depending upon process conditions. In addition, it was observed that an increase in L/S ratio and temperature might reduce the above concentration limit even more (see Figure 6b,c) to the level of approx. 8%, which is essential in reducing consumption of NaOH and the extent of aluminum contamination of the leachate. Molybdenum selectivity, in turn, was found to be negatively correlated in a statistically significant manner with three of the examined parameters: NaOH content, temperature, and L/S ratio, of which NaOH content was the most important. Those effects can be explained by extensive solubilization of aluminum at high NaOH concentrations, high temperatures, or with large amounts of NaOH (high L/S ratios). A more detailed analysis of the data shown in Figure 7a–c indicates that high molybdenum selectivity (of more than 90%) can be achieved in the examined system only at limited NaOH content, low-to-medium temperatures, and low L/S ratios, which conforms to the above-discussed correlations. Taking into account the above conclusions, the following conditions were selected for industrial-scale testing of the leaching operation: an NaOH content of less than 15%, a leaching temperature of not more than 60 °C, and with the minimum possible L/S ratio.

Industrial-scale molybdenum leaching tests were performed using the previously roasted catalyst of the composition shown in Table 5. Tests were performed in five batches of approx. 2500 kg each, and a total of 12,200 kg of the roasted catalyst was processed. Individual leaching experiments were performed for approx. 30 h, the time after which molybdenum content in the leachate stabilized. The process parameters mean values and their standard deviations are summarized in Table 7. A total volume of 35.2 m^3^ of the leachate, which contained 34.3 kg/m^3^ of molybdenum, was collected. The average molybdenum recovery achieved was a value of 87%, while aluminum recovery was 20%. The amounts of nickel and cobalt in the leachate were negligible.

### 3.3. Results of Precipitation of Molybdic Acid

The aquatic chemistry of molybdenum species strongly depends upon solution pH [23,52,53]. At high pH values (>5) the molybdate ion (MoO_4_^2−^) dominates, while in the solutions with slightly lower pH values the polymolybdate anion dominates, most likely heptamolybdate (Mo_7_O_24_^6−^). Those anions are known to produce highly soluble molybdenum salts. In contrast, much less soluble molybdic acid exists only at very low pH values. Analysis of the distribution of molybdenum species in a solution with 0.3 mol/L of molybdenum content, specific for the leachates tested in this work, indicated that the maximum fractions of molybdic acid and molybdenum trioxide (molybdic acid anhydrate) exist at a pH of around 0.25 (Figure 8b). However their amounts are very low, even at their maximum, and molybdic acid combined with molybdenum oxide constitutes only up to 3% of all the molybdenum species. The species that dominate at a pH of 0.25 are the cationic monomolybdenum (H_3_MoO_4_^+^ and HMoO_3_^+^) and dimolybdenum complexes (H_6_Mo_2_O_8_^2+^ and HMo_2_O_6_^+^), and an anionic polymolybdate anion (Mo_18_O_56_^4−^) as shown in Figure 8a. During the precipitation process, hydrolysis of the above cations can yield molybdenum oxide (from HMoO_3_^+^ and HMo_2_O_6_^+^), molybdic acid (from H_3_MoO_4_^+^) or dimolybdic acid (from H_6_Mo_2_O_8_^2+^). However, anionic polymolybdate (Mo_18_O_56_^4−^), which at a pH of 0.25 constitutes around 10% of total molybdenum compounds, needs to be converted further to the form of neutral acid or its cationic complexes, subject to hydrolysis. This can be achieved by a further reduction in solution pH to the level of 0.1 or less, at which negligible amounts of Mo_18_O_56_^4−^ exist (Figure 8a). Therefore, precipitation tests described further in this work were conducted at a pH of 0.0 ± 0.1 to maximize molybdic acid yield and to avoid formation of soluble molybdates.

Initial laboratory tests on precipitation of molybdic acid from the industrially obtained leachate were conducted by acidifying a 50 mL sample of these solutions with concentrated HCl to a pH of 0.0 ± 0.1. These tests were performed in triplicate, and the results are summarized in Table 8. It was found that although the solid product obtained herein was characterized by high molybdic acid content (>95%), the effectiveness of its precipitation was limited, and a low molybdenum recovery of around 56% was observed. This effect resulted from a high concentration of molybdenum in the filtrate, of approx. 7.6 g/L, thus there was a relatively high solubility of molybdenum compounds in the post-reaction mixture even at a pH of 0.0, possibly due to the presence of water-soluble complex molybdenum cations described previously. Therefore, we concluded that further efforts must be made to increase molybdic acid precipitation yield to the satisfactory level of around 95%, thus reducing molybdenum content in the supernatant solution to the value of less than 1 g/L. Our previous experiences on the precipitation of other d-block metal oxoacids indicate that the effectiveness of precipitation of these compounds can be significantly increased by adding ionic salts of ammonium or alkaline metals to the solution, as a result of the salting-out effect, which in our case would possibly reduce the stability of soluble molybdenum cationic complexes and increase the yield of their hydrolysis to molybdic acid. Therefore, the effects of the addition of ammonium chloride, sodium chloride, and potassium chloride on the solubility of molybdic acid were examined in separate laboratory experiments aimed at investigating the solubility of molybdic acid at a pH of 0.0 in the presence of these salts. Those experiments were described in detail in the experimental section of this work. As shown in Figure 9, the addition of each—NH_4_Cl, NaCl, and KCl—causes a significant decrease in the concentration of molybdenum in the supernatant solution with an increase in each salt concentration. The most substantial decrease in molybdenum content was observed for ammonium salt. In turn, the effects of the NaCl or KCl content were rather slim; thus, low molybdenum concentrations in the supernatant solution can be achieved only at a very high content of these salts. It should be mentioned that with the examined technology, sodium chloride is produced in the reaction mixture as a result of the neutralization of aluminates, excess NaOH, or directly as a co-product of molybdic acid precipitation reaction from the leachate with the following generalized reaction schemes:NaOH_(aq)_ + HCl_(aq)_ → NaCl_(aq)_ + H_2_O(13)
Na_2_MoO_4(aq)_ + 2 HCl_(aq)_ → H_2_MoO_4(s + aq)_ + 2 NaCl_(aq)_(14)

However, the NaCl amounts (equal to approx. 2 mol/L in our case) were insufficient to reduce the solubility of molybdic acid in the filtrate to 1 g/L or less (Figure 9), which would have facilitated high enough recovery of molybdic acid. In contrast to NaCl, ammonium chloride at a concentration of just 0.4 mol/L produced a supernatant solution with a molybdenum content of less than 0.4 g/L, suggesting that high recovery of molybdic acid can be achieved with the examined industrial leachate even when such low amounts of ammonium salts are present in the reaction mixture. Therefore, industrial tests on the precipitation of molybdenum from the examined leachate with hydrochloric acid were supplemented by the addition of concentrated ammonia to the reaction mixture, which resulted in the formation of ammonium chloride following the reaction:NH_3(aq)_ + HCl_(aq)_ → NH_4_Cl_(aq)_(15)

The amounts of hydrochloric acid and ammonia used were sufficient enough to produce just approx. 0.4 mol/L of NH_4_Cl in the reaction mixture while maintaining a solution pH of 0.0 ± 0.1.

Industrial-scale tests on the precipitation of molybdic acid were conducted using leachate samples produced previously with industrial tests on alkaline leaching. The results of these experiments are summarized in Table 8. A total volume of 32.0 m^3^ of the leachate was processed in six batches of approx. 5 m^3^ each. The filtrate contained just approx. 0.43 g/L of molybdenum, and molybdic acid was precipitated at a high molybdenum recovery of around 97.5%. A total of 1660 kg of yellow molybdic acid was produced with a purity of 97% and negligible amounts of aluminum, nickel, cobalt, and other potential contaminants. It should be mentioned that the solid product described herein was characterized by a much higher molybdic acid content than is commonly required for industrial molybdic acid (85% or more).

The XRD pattern of the obtained molybdic acid, presented in Figure 10, indicated a strongly crystalline structure of the precipitate, based on a high signal-to-background ratio, with orthorhombic molybdic acid H_2_MoO_4_ (ICDD PDF-2 #26-0149), and possibly monoclinic dimolybdic acid H_2_Mo_2_O_7_ (ICDD PDF-2 #37-0519) as major crystalline compounds of the product. The presence of dimolybdic acid may be explained by the hydrolysis of dimolybdenum cations (i.e., H_6_Mo_2_O_8_^2+^) produced by the acidification of the leachate to a pH of around 0.0, as discussed at the beginning of this chapter. Similarly, the presence of molybdic acid can be explained by the hydrolysis of monomolybdenum cations (i.e., H_3_MoO_4_^+^). The XRD pattern showed no evidence for molybdenum trioxide, MoO_3_, that potentially could have been produced by hydrolysis of HMoO_3_^+^ and HMo_2_O_6_^+^, probably because of the possibility of immediate hydration of MoO_3_ to molybdic acids in an aqueous solution.

## 4. Discussion

To evaluate the efficiency of converting molybdenum present in spent HDS catalysts to molybdic acid, recoveries for individual operations of the examined technology and total molybdenum recovery are summarized in Table 9. The total molybdenum recovery with the investigated technology equaled to 81%, which matches the molybdenum conservation criterion described in the introduction section of this work. This value can be considered to be satisfactory on the industrial scale, taking into account the complexity of the technology, its prolonged operation times, and the multi-stage nature of the operations conducted. Although precipitation and oxidizing roasting operations proceeded at relatively high molybdenum recoveries, each of around 95% or more, the alkaline leaching stage was characterized by a limited molybdenum recovery with an average value of just 87%, which led to the most significant loss of molybdenum among all of the operations considered, of approx. 13%. Such a limited recovery of molybdenum was likely the result of insufficiencies in the leaching of molybdenum from the roasted catalyst, with significant amounts of molybdenum remaining in the solid residue produced at this stage. However, it should be mentioned that the leached catalyst residue, when subjected to further hydrometallurgical treatment to recover other metals, such as nickel or cobalt, could generate additional amounts of molybdenum compounds which, when applicable, should have been converted to molybdic acid or other molybdenum compounds, thus increasing total molybdenum recovery. Furthermore, the determined molybdenum recovery for the leaching stage was characterized by a relatively high standard deviation of 8% due to inaccuracy in the processing of five separate batches, each of around 2500 kg of roasted catalyst, on the industrial scale. So, a high standard deviation suggests that with better control over leaching conditions, it should have been possible to improve the efficiency of this operation to the range of 90% or higher, which can produce an increase in total molybdenum recovery. Furthermore, the origin of the molybdenum losses during the oxidizing roasting operation, of around 4.3% by mass, was potentially the result of sublimation of the volatile MoO_3_, despite low roasting temperatures applied herein. In that case, since the flue gases produced by oxidizing roasting (such as SO_2_) are subject to alkaline scrubbing, as shown in Figure 1, molybdenum trioxide vapors can accumulate in scrubbing liquors as molybdates, and a strategy for their recovery as molybdic acid or other insoluble salts might be adopted, increasing overall molybdenum recovery. Finally, around 3% molybdenum loss during the precipitation stage was possibly caused by significant amounts of molybdenum remaining in the filtrate, of around 0.43 g/L (Table 10). Therefore, further removal of molybdenum compounds from this filtrate, which needs to be conducted to purify this waste regardless, could increase overall molybdenum recovery. It should also be mentioned that the amounts of molybdenum dissolved in the post-precipitation filtrate supports the necessity for low L/S ratio operation at a leaching stage, because higher L/S ratios used at this stage would result in larger volumes of the post-precipitation filtrate and more significant amounts of molybdenum remaining there, resulting in a reduction in molybdenum recovery.

## 5. Conclusions

It was demonstrated that it is possible to effectively recover molybdenum from exemplary spent hydrodesulfurization (HDS) catalysts in the form of molybdic acid (H_2_MoO_4_) on an industrial scale. The evaluated technology was based on three principal operations: oxidizing roasting of the spent catalyst with the formation of molybdenum trioxide (MoO_3_), leaching of the roasted catalyst with aqueous NaOH solutions leading to the formation of water-soluble sodium molybdate (Na_2_MoO_4_), and precipitation of molybdic (H_2_MoO_4_) acid with aqueous HCl from the previously produced leachate. It was found that molybdenum recovery for individual steps depends strongly upon process conditions. At the oxidizing roasting stage, molybdenum recovery depends primarily upon roasting temperature, which has to be limited because of the possibility of sublimation of MoO_3_ at high temperatures. In turn, it was found that molybdenum leaching from the roasted catalyst should have been carried out at low NaOH contents, low temperatures, and low liquid-to-solid (L/S) ratios to minimize co-leaching of aluminum along with molybdenum, and to maximize the molybdenum selectivity of the leaching stage. Finally, precipitation of molybdic acid from the leachate has to be conducted with an excess of hydrochloric acid at a pH of around 0 to mitigate the possibility of the formation of polymolybdates or similar anionic molybdenum complexes at high pH values.

In total, industrial-scale tests proved that with the above-described technology, it is possible to recover around 80% of molybdenum from the exemplary spent HDS catalyst in the form of a product, which contains around 97% H_2_MoO_4_. Molybdenum recoveries for the above-described unit operations, especially of the leaching stage, can be further improved by better control over process conditions at the industrial scale or recovery of molybdenum compounds from the wastes generated at every stage. The obtained product constitutes a tradeable commodity that can be freely sold to the market or converted to metal, other molybdenum compounds, or HDS catalysts. With an industrial-scale molybdenum recovery of around 80% and the potential for further enhancement in molybdic acid yield, the described technology matches the criterion for recycling technologies that facilitate conservation of molybdenum for further generations and could allow for the sustainable development of society with respect to the molybdenum economy.

## Figures and Tables

**Figure 1 materials-16-05762-f001:**
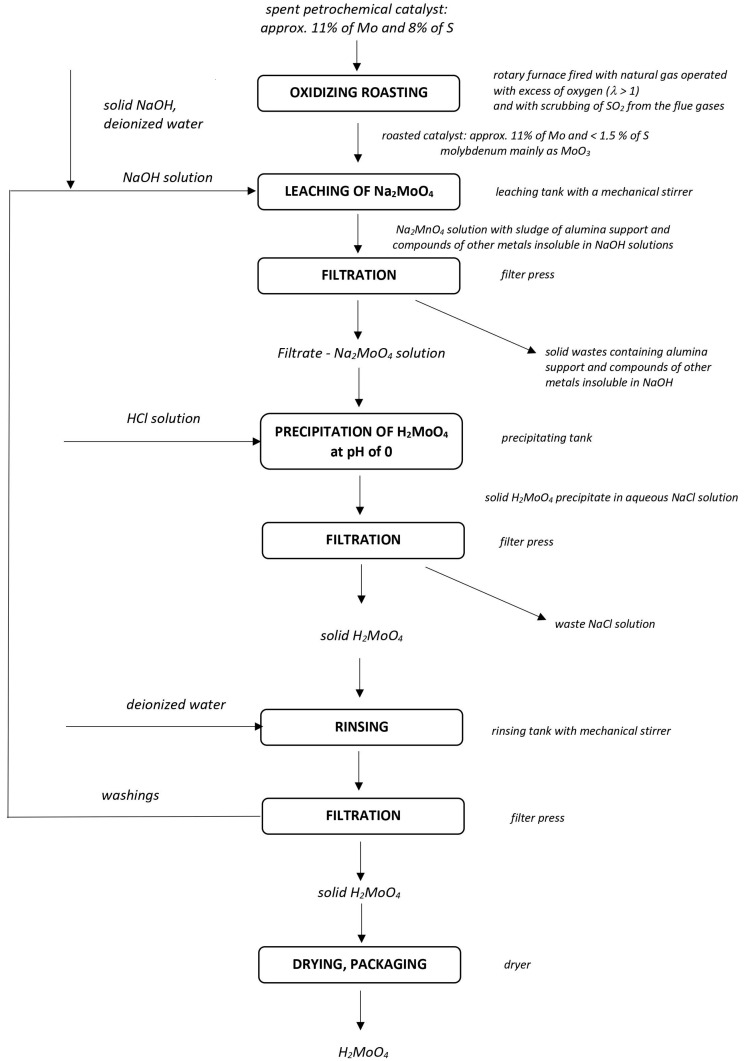
Schematic representation of the examined technology for recovery of molybdic acid from spent HDS catalyst.

**Figure 2 materials-16-05762-f002:**
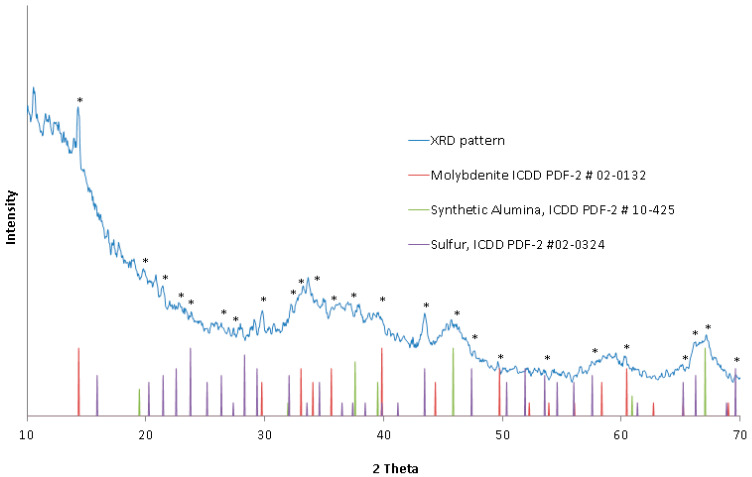
XRD pattern of the catalyst sample. Peaks that matched to the selected ICDD PDF-2 data files are marked with an asterisk.

**Figure 3 materials-16-05762-f003:**
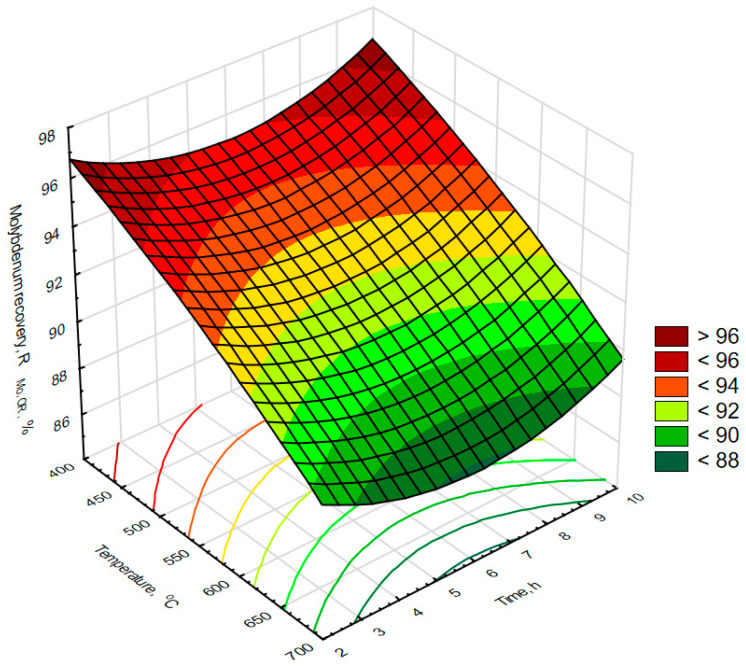
Effect of time and temperature on Mo recovery during laboratory-scale oxidizing roasting experiments. R^2^ for response surface type fit of the above presented data equals to 0.964.

**Figure 4 materials-16-05762-f004:**
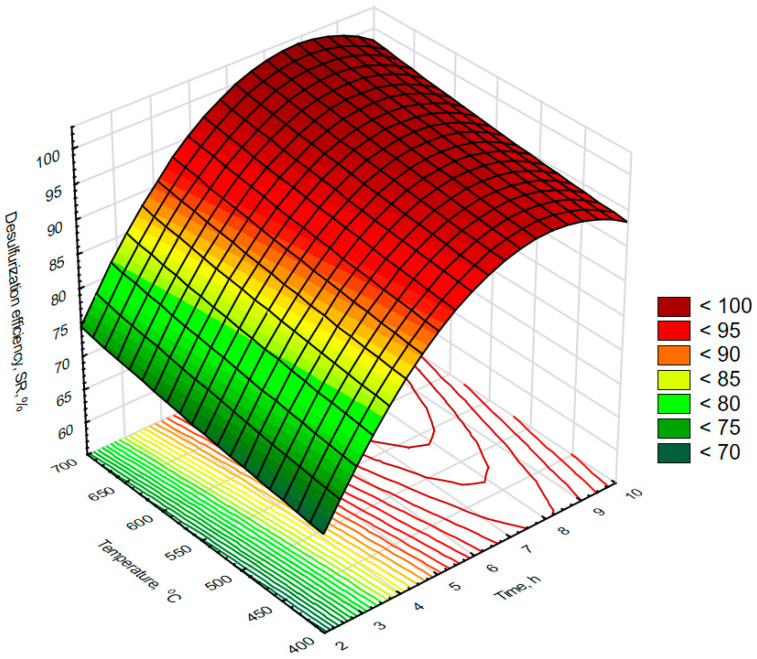
The effect of time and temperature on sulfur removal efficiency during laboratory-scale oxidizing roasting experiments. R^2^ for response surface type fit of the above presented data equals to 0.976.

**Figure 5 materials-16-05762-f005:**
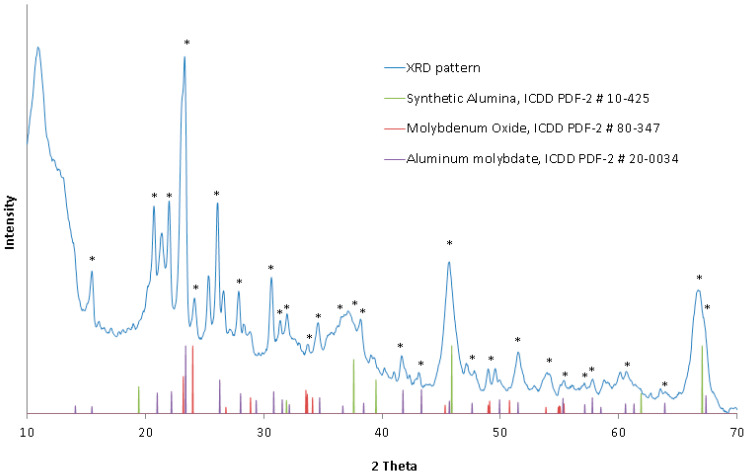
XRD pattern of the roasted catalyst. Peaks that matched to the selected ICDD PDF-2 data files are marked with an asterisk.

**Figure 6 materials-16-05762-f006:**
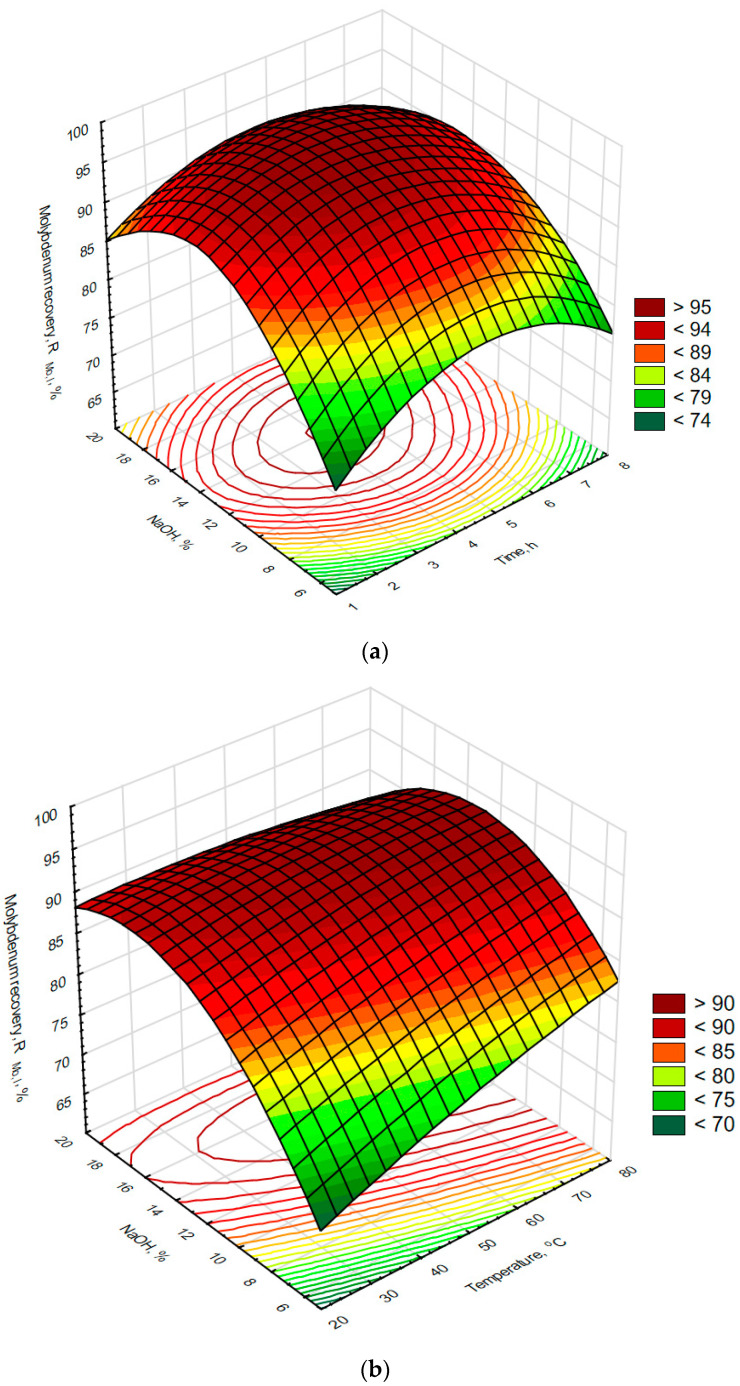
The effects of NaOH content and time (**a**), temperature (**b**), and L/S ratio (**c**) on recovery of molybdenum during alkaline leaching of the roasted spent HDS catalyst. R^2^ for response surface type fit of the above presented data equals to 0.834.

**Figure 7 materials-16-05762-f007:**
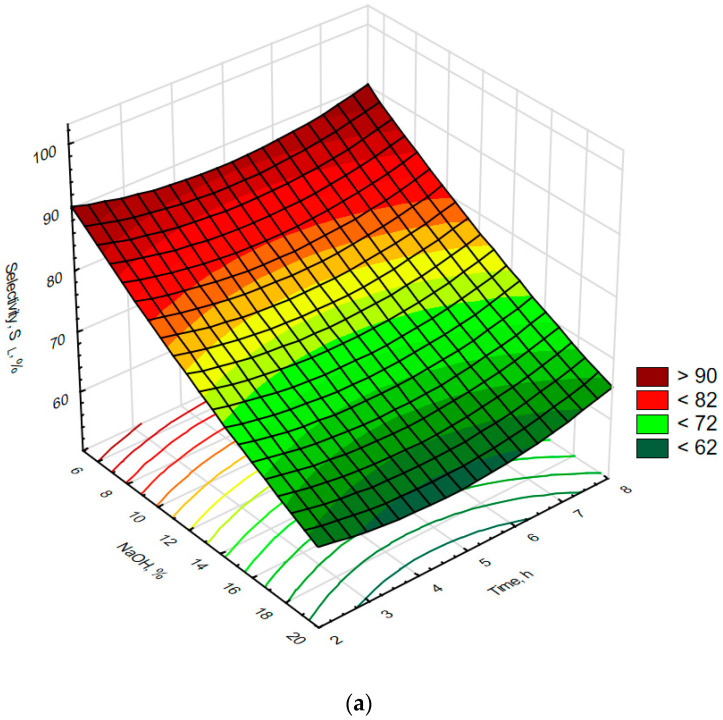
The effects of NaOH content and time (**a**), temperature (**b**), and L/S ratio (**c**) on selectivity of molybdenum recovery during alkaline leaching of the roasted spent HDS catalyst. R^2^ for response surface type fit of the above presented data equals to 0.966.

**Figure 8 materials-16-05762-f008:**
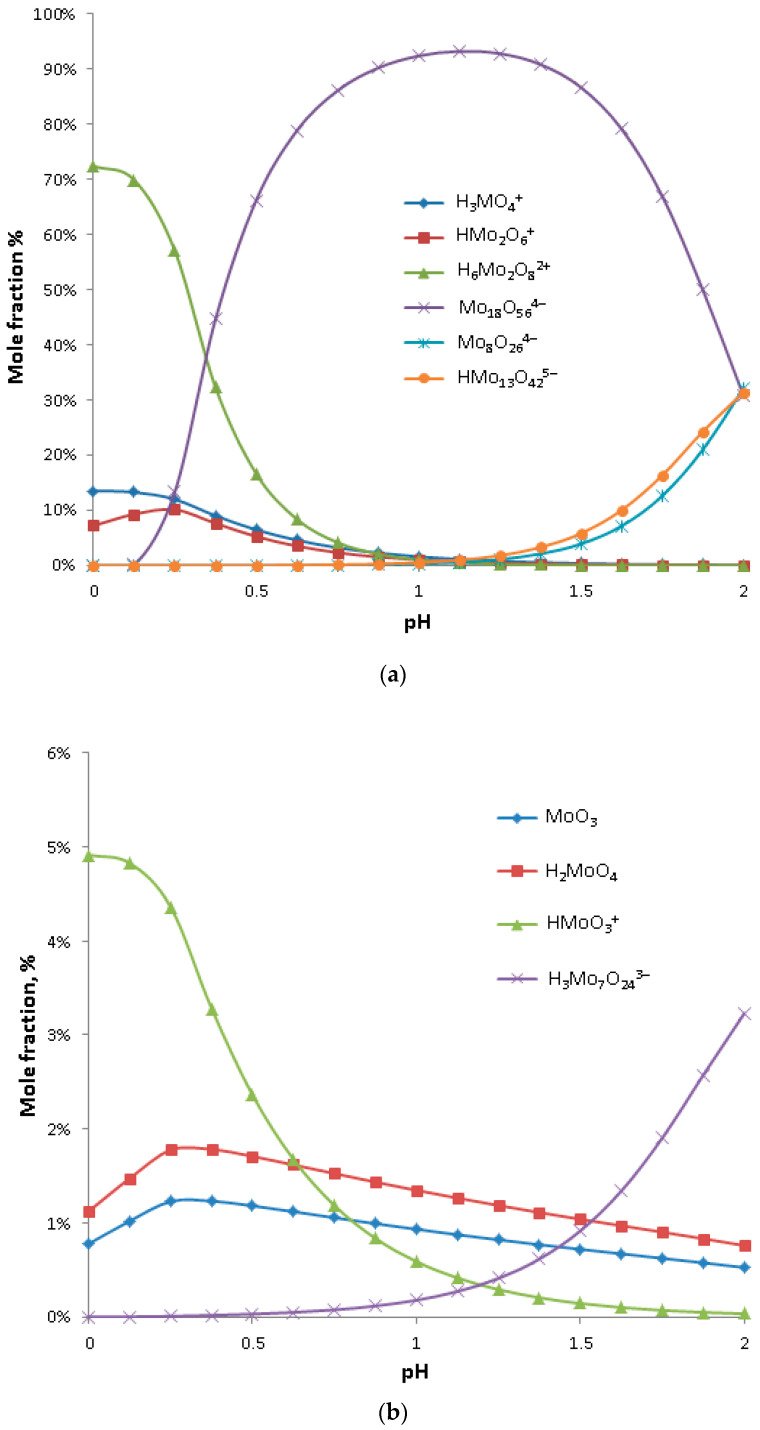
Distribution of aqueous molybdenum species at a pH range from 0 to 2 in a solution with total molybdenum content of 0.3 mol/L reproduced based on [53]. (**a**) presents distribution of major species, while (**b**) presents distribution of minor species.

**Figure 9 materials-16-05762-f009:**
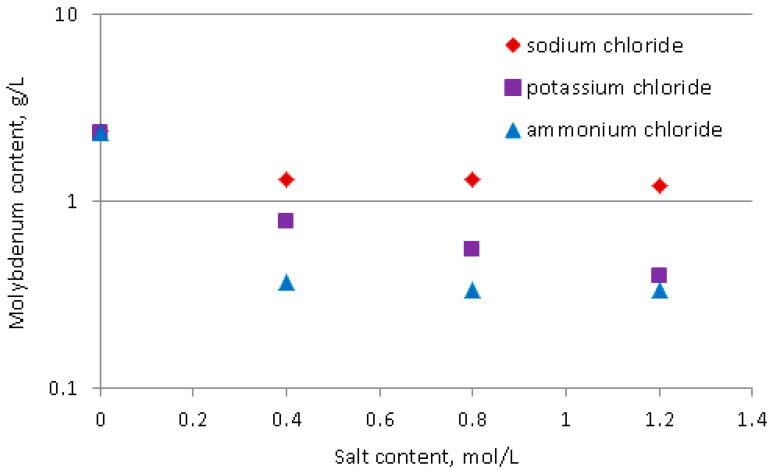
The effects of salt molar concentration on the concentration of molybdic acid in the supernatant solution at a pH of 0.0.

**Figure 10 materials-16-05762-f010:**
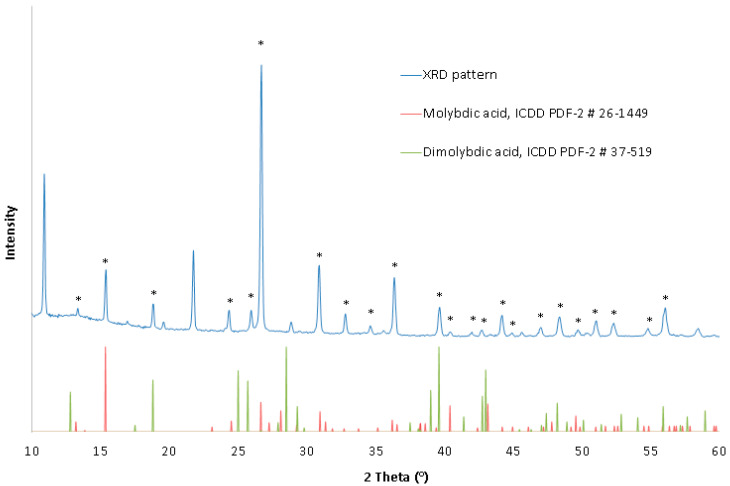
XRD pattern of the precipitated molybdic acid. Peaks that matched to the selected ICDD PDF-2 database records are marked with an asterisk.

**Table 1 materials-16-05762-t001:** Composition of the spent HDS catalyst tested in the laboratory.

Item	Mo	S	Al as Al_2_O_3_	Ni	Co
Mass percentage, %	11.6	7.71	25.0	4.72	1.66
Molar percentage, %	16.9	33.7	34.3	11.2	3.9

**Table 2 materials-16-05762-t002:** Descriptive statistics of experiments on the roasting of spent petrochemical catalysts.

Variable	No. of Experiments	Average	Min.	Max.	Standard Deviation
Time, t	12	5.66	2.0	10.0	3.60
Temperature, T	12	537.5	400	700	135
Sulfur removal efficiency, SR	12	86	68	99	13
Mo recovery, R_Mo,or_	12	93.1	88.2	97.2	3.4

**Table 3 materials-16-05762-t003:** Matrix of correlation coefficients (R) for oxidizing roasting experiments. Statistically significant correlation coefficients (*p* < 0.05) are shown in bold.

Variable	Statistics		Correlation Coefficients (R)			
	Average	Standard Deviation	t	T	SR	R_Mo,or_
time, t	5.67	3.60	1.00	−0.12	0.85	0.09
Temperature, T	537.5	135	−0.12	1.00	0.09	−0.95
Sulfur removal efficiency, SR	85.8	12.7	0.85	0.09	1.00	−0.21
Mo recovery, R_Mo,or_	93.1	3.37	0.09	−0.95	−0.21	1.00

**Table 4 materials-16-05762-t004:** Results of industrial scale roasting of spend HDS catalyst.

Item	Mass, kg	Mo, % (m/m)	S, % (m/m)	Al as Al_2_O_3_, %	Ni + Co, %
Spent HDS catalyst	25,600	10.6 ± 1.4	8.6 ± 1.4	24.1 ± 0.9	5.8 ± 0.9
Roasted catalyst	20,950	12.4 ± 1.4	1.5 ± 0.7	29.5 ± 1.4	7.0 ± 1.2

**Table 5 materials-16-05762-t005:** Descriptive statistics of experiments on the leaching of roasted petrochemical catalysts.

Variable	Number of Experiments	Average	Minimum	Maximum	Standard Deviation
time, t	25	4.5	1.0	8.0	3.2
NaOH, %	25	12.95	5.00	20.00	7.03
Temperature, °C	25	50.0	20.0	80.0	27.4
L/S	25	5.00	3.00	7.00	1.83
R_Mo,l_, %	25	83.1	60.0	94.7	9.4
R_Al,l_, %	25	15.3	0.3	37.7	9.9
S_l_, %	25	85.6	71.1	99.6	8.0

**Table 6 materials-16-05762-t006:** Matrix of correlation coefficients (R) for alkaline leaching experiments. Statistically significant correlation coefficients (*p* < 0.05) are shown in bold.

	Statistics		Correlation Coefficients						
	Average	Standard Deviation	t, h	NaOH, %	T, °C	L/S	R_Mo,l_, %	R_Al,l_, %	S_l_, %
t, h	4.50	3.20	1.00	0.049	−0.050	−0.050	0.149	0.164	−0.095
NaOH, %	13.0	7.04	0.049	1.000	0.049	0.049	**0.655**	**0.748**	**−0.914**
T, °C	50.0	27.4	−0.050	0.049	1.000	0.050	0.381	**0.552**	−0.247
L/S	5.00	1.83	−0.050	0.049	0.050	1.000	0.259	0.276	−0.263
R_Mo,l_, %	83.4	8.65	0.149	**0.655**	0.381	0.259	1.000	**0.640**	**−0.752**
R_Al,l_, %	15.8	7.91	0.164	**0.748**	**0.552**	0.276	**0.640**	1.000	**−0.887**
S_l_, %	76.6	13.9	−0.095	**−0.914**	−0.247	−0.263	**−0.752**	**−0.887**	1.000

**Table 7 materials-16-05762-t007:** Summary of the industrial-scale leaching of the roasted catalyst with NaOH solutions.

Item	Value
Roasted catalyst mass, kg	12,200
Total mass of deionized water, kg	33,000
Total mass of NaOH, kg	4700
NaOH content, %	12.9 ± 2.6
Temperature, °C	55 ± 5
L/S	2.7 ± 0.3
Leachate total volume, m^3^	35.2
Leachate molybdenum content, kg/m^3^	34.3 ± 5.6
Leachate aluminum content, kg/m^3^	19.8 ± 6.2
Leachate nickel content, kg/m^3^	<0.050
Leachate cobalt content, kg/m^3^	<0.050
Molybdenum recovery, %	87 ± 8
Aluminum recovery, %	20 ± 6

**Table 8 materials-16-05762-t008:** Summary of laboratory results of precipitation of molybdic acid from the industrial leachate with concentrated HCl.

Item	Concentration
Filtrate molybdenum content, g/L	7.6 ± 0.9
Molybdenum recovery (R_Mo,p_), %	56 ± 2

**Table 9 materials-16-05762-t009:** Summary of molybdenum recovery in the tested technology.

Step	Overall Molybdenum Recovery, %	Molybdenum Loss, %
Oxidizing roasting	95.7 ± 1.6	4.3 ± 1.6
Alkaline leaching	87 ± 8	13 ± 8
Precipitation of molybdic acid	97.5 ± 0.6	3.0 ± 0.6
Total	81 ± 9	19 ± 9

**Table 10 materials-16-05762-t010:** Summary of industrial scale molybdic acid precipitation conditions and results.

**Item**	**Value**
Total leachate volume, m^3^	32
Total volume of concentration HCl, m^3^	16.0
Total volume of concentration NH_3_, m^3^	3.0
Molybdenum content in the filtrate, g/L (kg/m^3^)	0.43 ± 0.13
NH_4_Cl content in the supernatant solution, mol/L	0.39 ± 0.05
Molybdenum recovery, %	97.5 ± 0.6
Mass of dry molybdic acid, kg	1660
Molybdic acid content, %	97.0 ± 0.6
Aluminum content, %	<0.14
Nickel content, %	<0.1
Cobalt content %	<0.1
Sodium content, %	<0.1
Chloride content, %	<0.1

## Data Availability

Data not available due to privacy.

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
