# Peer review of "Industrial-Scale Technology for Molybdic Acid Production from Waste Petrochemical Catalysts"

_materials, 2023, doi:10.3390/ma16175762_

Round 1

Reviewer 1 Report

The paper addresses an industrially important topic of Mo recovery from spent catalysts. Existing literature describing the fundamental concepts and options of recovery processes is critically evaluated and the process design is made based on the literature review. While no fundamentally new concepts are introduced in the present paper, it is focused on evaluating the designed process performance in larger scale, which was not published before, and this aspect ensure the novelty of the paper. The paper contains valuable data which justify publication in Materials, but in some parts their presentation is inadequate (as specified below in detail). Therefore I recommend authors to revise their manuscript accordingly.

L159 mi – initial mass of the roasted catalyst – it should probably refer to initial mass of raw catalyst

L159, 208, 245 – the introduced recovery levels R should be named as partial, because they refer to only the roasting, leaching, or precipitation step respectively

L307 – RMo,ox symbol should probably match the RMo,or introduced in L157

L306 – The text probably refers to tables 2 and 3 instead of table 4 or tables 2 and 3 are not referenced at all. Authors should carefully check the document for those formal aspects as the present state does not support easy orientation in the text

L357 – The speculation about the laboratory/industrial process duration related to the ratio between the flue gas rate, lambda and catalyst mass should be supported by rough calculation which is possible, as the flue gas rate, lambda and catalyst mass are all known

Fig. 3 and 4 display smoothed profiles based on the 12 experiments suggested in tables 2 and 3. There is no information on the goodness of fit of the smoothed surface. I suggest to include detailed setups and results of all individual experiments and possibly remove the statistic overview in tables 2 and 3. The objective is to observe trends, so using descriptive statistics is of no use and those trends are obviously not linear, so that the correlation coefficients are also of limited information value. The same objection is for presenting the results of leaching. On the other hand the precipitation data are presented in scientifically sound manner.

L398 – The leaching experiments used the industrially roasted catalyst. I just want to suggest that testing the leaching performance of laboratory roasted catalysts (at different conditions) could be interesting. The authors should justify their decision not to explore this dependence.

Small typographic errors are abundantly present throughout the paper (comas, punctuation, ° characters, spacing in the formulae, symbol italics, …). Either authors or the editorial office should work on this before publication

Author Response

We would like to thank you for taking the time to review our article and for your valuable comments.  

We have adopted all suggestions in the revised manuscript as follows:  

  • L159 corrected as requested,  
  • L159, 208, 245 corrected as requested. Moreover, in table 10 we emphasized that molybdenum recover presented there describe molybdenum recovery for the technology as a whole, 
  • in lines 159, 208 and 245 we emphasized that the calculated yields are partial and relate to a given stage of technology,  
  • L307 corrected as requested, 
  • L306 – cited references were corrected as requested, 
  • L357 – we agree that roasting time can be theoretically calculated for the industrial scale, however, our experience show that on the industrial scale, real times are much longer than those calculated with the theoretical formulas, possibly due to the heat loss or mass exchange inefficiencies of the roasting process. Therefore we opt to provide the readers with more accurate but realistic roasting times for our furnace, which was introduced in the revised manuscript, 
  • Figs 3, 4, 6 and 7 following the reviewer comments we provided correlation coefficients (R2) for response surface fits used to present those data which, in our opinion, will provide sound information on the goodness of fit of our data to the tendencies which are clearly visible in the figures. We would also prefer to save descriptive statistics analysis since they show general correlations that exist in the examined systems, 
  • L398 – we opted to test leaching conditions using industrially roasted catalyst samples rather than laboratory samples because they resemble the overall industrial-scale operation results much closer with maximum molybdenum oxidation and sulfur removal efficiency during the roasting stage, which was achievable on industrial scale rather than in the laboratory. In our opinion, conducting those tests based on laboratory-roasted catalyst samples will introduce uncertainty in the developed technology related to scaling up from the laboratory to industrial. 

Best regards, 

Ewa SzydÅ‚owska-Braszak 

Reviewer 2 Report

The authors present the results of well-done study, and the method for Mo recovery proposed by them was successfully tested in industrial scale. I recommend to publish the manuscript almost in its present form. One minor comment: the authors cite PDF base when they assign reflections on powder XRD patterns (page 8, 12, 22). I recommend to cite original papers, which were used for creation of the respective PDF cards.

Author Response

We would like to thank you for taking the time to review our article and for your valuable comments.  We have adopted all suggestions in the revised manuscrip.

Reviewer 3 Report

The “materials-2540727” manuscript describes a very interesting work for recovering Mo from spent HDS catalysts. Laboratory and Industrial scale experiments have been organized and performed. They concern a three-stage treatment of spent catalyst (roasting, leaching, and precipitation). Optimization of the conditions adopted in each step has been performed based on the laboratory experiments. The efficiency of the proposed process is well proved in industrial scale. Thus, I propose publication after minor revision according with the following comments.

Figure 1: Please check this figure and change Mn by Mo, where it is necessary.

Table 3: It is difficult to be read. Please, pay attention to its appearance (where are the bold?) and for the correct writing of some subscripts.

Figure 8a: Please, check the formulas.

Figure 10. The figure caption should be corrected. The XRD pattern presented concerns the precipitate obtained. Some intense peaks have not been assigned and the intensities of some other are not analogous with those suggested by the database cards. Have the authors any explanation?

Conclusions: The influence of ammonia addition should be mentioned.

Author Response

We would like to thank you for taking the time to review our article and for your valuable comments.

We have adopted all suggestions in the text with as follows:

- Figure 1 - corrected as requested,

- Table 3 - edited and corrected as requested,

- Fig 8a – we checked all the formulas in Fig 8a, they match data presented in Table 1 of ref [53] and thus they are correct,

- Fig 10 - figure caption was corrected as requested. Some of the long lines couldn’t be matched to any compounds possibly due to the poor quality of the sample, however, the data in figure best match to molybdic and dimolybdic acid we could achieved.

Best regards,

Ewa Szydłowska-Braszak

Round 2

Reviewer 1 Report

All crucial issues have been solved. Authors decided not to include full presentation of primary data, but at least they added goodness of fit to figure captions, so that reader can get impression about the data variability. The paper can be published in Materials.